# Characterization of the Complete Mitochondrial Genome of the Elongate Loach and Its Phylogenetic Implications in Cobitidae

**DOI:** 10.3390/ani13243841

**Published:** 2023-12-13

**Authors:** Zhenlin Ke, Kangqi Zhou, Mengdan Hou, Hui Luo, Zhe Li, Xianhui Pan, Jian Zhou, Tingsen Jing, Hua Ye

**Affiliations:** 1Integrative Science Center of Germplasm Creation in Western China (Chongqing) Science City, Key Laboratory of Freshwater Fish Reproduction and Development (Ministry of Education), Key Laboratory of Aquatic Science of Chongqing, College of Fisheries, Southwest University, Chongqing 402460, China; 13022365884@163.com (Z.K.); hou660x@163.com (M.H.); luohui2629@126.com (H.L.); jingtingsen@163.com (T.J.); 2Key Laboratory of Aquatic Science of Chongqing, Chongqing 400175, China; 3Guangxi Key Laboratory of Aquatic Genetic Breeding and Healthy Aquaculture, Guangxi Academy of Fishery Sciences, Nanning 530021, China; zhoukqfisher@163.com (K.Z.); lzyzy0515@163.com (Z.L.); 15002381261@163.com (X.P.); 4Fisheries Institute, Sichuan Academy of Agricultural Sciences, Chengdu 611731, China

**Keywords:** elongate loach, Cobitidae family, mitochondrial structural characteristics, phylogenetic analysis

## Abstract

**Simple Summary:**

The complete mitochondrial genome has been widely used in phylogenetics-related studies, as it offers valuable insights into evolutionary relationships. In this study, we reported the complete mitogenome of the elongate loach (*Leptobotia elongata*) and conducted a detailed analysis of its characteristics which was employed to infer phylogenetic relationships. These findings reveal that both the gene arrangement and composition of mitochondrial genes in the elongate loach are comparable to those found in other bony fishes. Our study further demonstrated that the Cobitidae species under investigation could be grouped into two distinct clades, with elongate loach showing a sister relationship with *L. microphthalma*. Collectively, our research enhanced the understanding of the mitochondrial genome structure and contributed to the phylogenetic analysis of the elongate loach.

**Abstract:**

The elongate loach is an endemic fish in China. Previous studies have provided some insights into the mitochondrial genome composition and the phylogenetic relationships of the elongate loach inferred using protein-coding genes (PCGs), yet detailed information about it remains limited. Therefore, in this study we sequenced the complete mitochondrial genome of the elongate loach and analyzed its structural characteristics. The PCGs and mitochondrial genome were used for selective stress analysis and genomic comparative analysis. The complete mitochondrial genome of the elongate loach, together with those of 35 Cyprinidae species, was used to infer the phylogenetic relationships of the Cobitidae family through maximum likelihood (ML) reconstruction. The results showed that the genome sequence has a full length of 16,591 bp, which includes 13 PCGs, 22 transfer RNA genes (tRNA), 2 ribosomal RNA genes (rRNA), and 2 non-coding regions (CR D-loop and light chain sub-chain replication origin OL). Overall, the elongate loach shared the same gene arrangement and composition of the mitochondrial genes with other teleost fishes. The Ka/Ks ratios of all mitochondrial PCGs were less than 1, indicating that all of the PCGs were evolving under purifying selection. Genome comparison analyses showed a significant sequence homology of species of Leptobotia. A significant identity between *L. elongata* and the other five Leptobotia species was observed in the visualization result, except for *L. mantschurica*, which lacked the tRNA-Arg gene and had a shorter tRNA-Asp gene. The phylogenetic tree revealed that the Cobitidae species examined here can be grouped into two clades, with the elongate loach forming a sister relationship with *L. microphthalma*. This study could provide additional inferences for a better understanding of the phylogenetic relationships among Cobitidae species.

## 1. Introduction

The elongate loach (*Leptobotia elongata*), belonging to Cobitidae of Cypriniformes, is indigenous to the middle and upper reaches of the Yangtze River in China [1]. It is characterized by rapid growth and exceptional ornamental value [2,3]. However, the wild population resources of the elongate loach have experienced a significant decline since the 1980s due to overfishing, dam construction, and destruction of feeding and spawning grounds [4]. As a result, it has been classified as vulnerable grade (VU) in the China Red Book of Endangered Animals-Fish [5]. 

The family Cobitidae was originally proposed by Regan [6]. In this family, extensive research focused on morphological characteristics and mitochondrial genes has been conducted for over a century [7,8,9,10,11]. Currently, many scientists tend to divide Cobitidae into three subfamilies: Nemacheilinae, Botiinae, and Cobitinae [12]. In order to maintain consistency between the phylogenetic relationship and the natural classification of Cobitidae fishes, Tang et al. [13] elevated these three subfamilies to the family level, which aligns with the classification of Liu et al. [9]. As the second-largest group of Cypriniformes, Cobitidae is a key element in resolving the phylogenetic relationships of Cypriniformes. Investigating the phylogenetic relationships of the elongate loach, one of the youngest species in the Cobitidae, is beneficial to resolve the taxonomic ambiguity of Cobitidae fishes. Previous studies only focused on biological characteristics [14], artificial breeding [15], embryo development, and genetic diversity [16,17]. However, its research on the phylogenetic relationships of *L. elongate* is limited [18]. Studies dealing with the phylogenic status of the elongate loach addressed questions about inter-family relationships in Cobitidae [6,11,12,13], but research into species phylogenetic relationships within the family remain lacking. Therefore, a reevaluation of the phylogenetic relationships of the elongate loach, involving additional genes and a broader range of species, could provide more data for the conservation of the elongate loach’s wild population resources.

Mitochondrial DNA (mtDNA) is present in the cells of all eukaryotes [19]. Compared to nuclear genes, mtDNA evolves at a faster rate, allowing for a more accurate representation of phylogenetic relationships. Therefore, mtDNA is widely utilized as a molecular marker in phylogenetic studies [20,21,22,23]. In fish phylogeny research, genes such as cytochrome b (*cytb*), cytochrome oxidase (*cox*), and 16 S rRNA are commonly employed at the species-to-family level [22,24,25]. However, relying solely on a single mitochondrial gene may lead to misleading phylogenetic data due to limited information capacity and homogenization effects [26]. In contrast, utilizing the complete mitochondrial genome could provide a more comprehensive set of phylogenetic information [27].

In this study, we sequenced the mitochondrial genome, analyzed the structural information of the elongate loach, and compared the structures and complete mitochondrial genome with some of the determined Leptobotia species. Additionally, we reconstructed phylogenetic trees using complete mitochondrial genome sequences to analyze the evolutionary relationships of the elongate loach in the Cobitidae family. This study might provide further insight into the structure of *L. elongate* and improve our understanding of the evolutionary relationships of *L. elongate* which will be helpful to resolve uncertainties within the Cobitidae family. 

## 2. Materials and Methods

### 2.1. Sample Collection and DNA Extraction

The elongate loach is adorned with a combination of brown and yellow hues throughout its body. Its head and sides are embellished with irregular spots of various shades of color. There are striking black stripes in the elongate loach. The smaller eyes are in the lateral upper position, and the horseshoe-shaped fissure is in the lower position with two pairs of kissing whiskers and one pair of mouth beards (Appendix A). An elongate loach sample was collected from a breed pond at the Sichuan Fisheries Research Institute of Chengdu (103°54′53.740″ E, 30°45′26.956″ N), Sichuan Province, China, in October of 2020. The pectoral fin of one elongate loach was collected and stored in 95% ethanol at −20 °C. Genomic DNA was isolated from the pectoral fin using the phenol-chloroform method and the quality and integrity of DNA samples were assessed using an Agilent 2100 Bioanalyzer 2.2. Mitochondrial genome sequencing and assembly.

After qualifying the DNA sample, the DNA was mechanically fragmented using ultrasonic interruption. The fragmented DNA underwent fragment purification, end-repair, addition of A at the 3′ end, connection of sequencing adapters, and selection of fragments of different sizes using agarose gel electrophoresis. PCR amplification was then performed to generate a sequencing library [18]. The qualified PCR products were sequenced on the Illumina HiSeq 2500 platform. 

Prior to assembly, low-quality data, including the reads of average quality value < 5 or N content > 5, were filtered using Fasta software (version 0.20.0), and the sequences linker and primer sequence were trimmed from the reads. The mitochondrial genome assembly was carried out using the following methodology. First, clean reads were assembled using SPAdes (version 3.10) [28] to obtain SEED sequences, and the seed sequences were iterative extended in GCE (version 1.0.2) to obtain contig sequences. Second, contigs were connected to generate scaffold sequences using SSPACE (version 2.0 (https://www.baseclear.com/services/bioinformatics/basetools/sspace-standard/, accessed on 8 September 2022), and Gaps in the scaffold sequences were filled using Gapfiller (version 2.1.1 (https://sourceforge.net/projects/gapfiller/, accessed on 8 September 2022) until a complete pseudo genome sequence was assembled. Lastly, the sequencing results were mapped onto the assembled pseudo genome sequence to identify and correct any incorrect bases, and the complete mitochondrial circular genome sequence was obtained by coordinate remaking.

### 2.2. Mitochondrial Genome Annotation and Analysis

The newly assembled sequences were annotated in the Mitos web server (http://mitos2.bioinf.uni-leipzig.de, accessed on 8 September 2022) [29] with the following parameters: E-value Exponent = 5, Maximum Overlap = 100, ncRNA overlap = 100. The annotation results were then compared with those of closely related species. Finally, after manual correction, the final annotation results were obtained.

The secondary structure of tRNAs was obtained from the annotation results. The circular map of the mitochondrial genome was generated using OGDRAW (version 1.3.1) [30]. The relative synonymous codon usage (RSCU) values were analyzed with MEGA (version 6.0). The mitochondrial genome skew values were calculated using the following formula: ATskew = (A − T)/(A + T); GCskew = (G − C)/(G + C) [31]. Mafft (version 7. 310) [32] software was used for gene sequences comparison between the elongate loach and six Leptobotia fishes (*L. mantschurica*, *L. taeniops*, *L. microphthalma*, *L. rubrilabris*, *L. punctata*, and *L. pellegrini*), and the evolutionary rate (Ka/Ks, ω) was calculated using KaKs_Calculator (version 2.0) [33]. If the evolutionary rate is equal to 1, >1, or, <1, the PGCs are expected to be under no selection, positive selective constraint (purifying selection), or diversifying selection [34]. The mitochondrial genome structure was compared between the elongate loach and six Leptobotia fish species in a CGVIEW server [35] with default parameters (http://stothard.afns.ualberta.ca/cgview_server/, accessed on 12 September 2021) and the alignment results were visualized using mauve software (version 2.4.0).

### 2.3. Phylogenetic Analyses

The phylogenetic tree was reconstructed using the complete mitochondrial genome sequences of 36 Cypriniformes species, with *Myxocyprinus asiaticus* and *Danio rerio* used as outgroups (Table 1). All the genome sequences were set to the same start points in the circular sequence. Multiple sequence alignment was performed in MAFFT software (version 7.42) with auto model, and the alignment sequences were trimmed using trimAl (version 1.4. rev15). Subsequently, the RaxML (version 8.2.0) software was used to conduct the rapid bootstrap analysis (bootstrap = 1000) to construct the maximum likelihood evolution tree. 

## 3. Results and Discussion

### 3.1. Mitochondrial Structural Characteristics 

The complete mitochondrial genome of the elongate loach was obtained through high-throughput sequencing technology (OR818399), with a total length of 16,591 bp (Figure 1). It consists of 37 typical animal mitochondrial genes, including 22 tRNA genes, 13 PCGs, 2 rRNA genes, and 2 non-coding regions (D-Loop and OL). Among the mitochondrial genes, nine genes (*trnQ*, *trnP*, *trnE*, *nad6*, *trnS2*, *trnY*, *trnC*, *trnN*, *trnA*) were encoded by the light (L) strand, while the remaining genes were encoded by the heavy (H) strand. The arrangement and content of the mitochondrial genome in the elongate loach were similar to those reported in teleost fishes [20,36,37]. The entire base composition of the elongate loach mitochondrial genes is as follows (Table 2): 30.79% A, 24.77% T, 16.17% G, and 28.27% C, and the AT and GC percentages are 55.56% and 44.44%, respectively, which results in a positive skew value for AT and a subtractive skew value for CG. It was suggested that the occurrence of A and C bases was more frequent in the genome. Previous studies have shown that the bias in base composition plays a crucial role in the replication and transcription of mitochondrial genomes [38].

### 3.2. Protein Coding Genes

The PCGs account for 68.89% of the total length of the elongate loach mitochondrial genome. As expected (Table 3), most PCGs started with the regular codon ATG, except for the *cox1* which started with GTG. Among the PCGs, there were seven genes that shared the complete stop codon TAA, while six genes shared incomplete stop codons (TA- or T--) which exists in many teleosteans as shown in numerous studies: *L. microphthalma* with seven incomplete stop codons [39], *Cobitis macrostigma* with seven incomplete stop codons [40], *Pelteobagrus fulvidraco* with five incomplete stop codons [20], *Parabotia kiangsiensis* with three incomplete stop codons [41], etc. The presence of tRNA sequences at the 3’ end of these genes is responsible for the incomplete stop codons [42], and these incomplete stop codons can be converted to TAA through post-transcriptional polyadenylation [43]. 

Three overlapping regions between certain PCGs (ATPase8-ATPase6, ND4-ND4L, and ND5-ND6) were also identified in this study. These overlapping regions were 4–10 bp in length, with the largest overlapping occurring between ATP8 and ATP6, which was common among Cobitidae species [44]. These overlapping regions contribute to the variation in mitochondrial genome length among closely related species [45]. The relative synonymous codon usage (RSCU) values of PCGs are revealed in Table 4 and Figure 2. In the protein-coding region, a total of 2012 codons were used. According to the degeneracy of codons, serine and leucine were encoded by six codons, while the remaining amino acids were encoded by either four or two codons. In the coded passwords, CUA (leucine), AUU (isoleucine), GCC (Aminopropanoic), and GCA (Aminopropanoic) are the most common, while AAA (Lysine) and CUA (leucine) have the highest RSCU values. Therefore, PCGs preferred the codons using adenine at the third codon. The codon usage varied between different species, which was more prominent between species with a more distant evolutionary relationship [46]. It is relevant to gene length, mutation bias, GC composition, amino acid composition, tRNA abundance, and translational selection [47,48,49,50,51,52]. 

### 3.3. Genome Comparative Analysis

The nonsynonymous substitution ratio (Ka) and synonymous substitution ratio (Ks) were calculated to evaluate selective pressures during the evolutionary process of PCGs among Leptobotia species. It was shown that the average Ka was similar among the six fishes (0.0089–0.0114), with *nd5* exhibiting the highest average Ka (Figure 3A; Appendix A), indicating that it might be under positive selection across species. The Ks of *L. microphthalma* was significantly lower than the other species (Figure 3B; Appendix A). There were more synonymous substitutions per synonymous sites in *nd*4 and *atp6*, exhibiting the high polymorphic nature of the genes in these fishes. *nd4* has also been confirmed to be polymorphic among sharks [53] and blue-spotted maskray [45]. The Ka/Ks ratio (ω) is a means to examine molecular adaption [54,55], which could be used to estimate the evolutionary rate among Cobitidae species. In this study, the Ka/Ks ratios of all PGCs were less than 1, indicating that purifying selection possesses the leading role in the evolution of these PGCs (Figure 3C; Appendix A). Therein, *cox3* (0.0076) and *nd4l* (0.0087) were evolving under a strong purifying selection, whereas *nd4* (0.0549), *nd5* (0.0782), and *nd2* (0.0784) were evolving under comparatively relaxed mutational constraints. Currently, selective pressure in mitochondrial PCGs has been poorly studied in other Cobitidae species [13,18,39,40,56,57,58], while the same pattern of widespread purifying selection has been discovered in several other decapod crustaceans [59]. 

The comparison of the mitochondrial genome sequences between the elongate loach and six Leptobotia species showed a significant sequence homology within the Leptobotia genus (Figure 4 and Figure 5). The elongate loach showed a higher identity with the other five species, except for *L. microphthalma*, which lacked the *tRNA-Arg* and a shorter *tRNA-Asp*, indicating that the arrangement of genes of the Leptobotia species is comparatively conserved.

### 3.4. Ribosomal RNA and Transfer RNA Genes

The total length of rRNAs was 2638 bp, with an AT skew value of 0.272 and a GC skew value of −0.095. The lengths of 12 S rRNA and 16 S rRNA were 955 bp and 1683 bp, respectively (Table 3). These rRNAs were located between *tRNA-Phe* and *tRNA-Leu* and were separated by *tRNA-Val*, which is consistent with the most reported teleost [60].

There were 22 tRNAs in the mitochondrial genome of the elongate loach, with a total length of 1558 bp. The AT content was 53.89% and the AT skew value was 0.044. Each tRNA has a length of 66–76 bp, with 14 encoded in the H chain and 8 encoded in the L chain. Most of the secondary structure of tRNA genes (Figure 6) in the elongate loach were standard clover-shaped, except for *trnS1*, which lacked the DHU stem. It was very common to defect DHU stem in metazoan [43]. Additionally, there were 18 false GU pairings in the tRNA sequences of the elongate loach. GU mismatch was frequently observed in teleost fishes and allowed for an expanded chemical and conformational diversity of double-stranded RNA. This diversity provided unique sites that were recognized by amino acids, contributing to a higher genetic diversity for the elongate loach [61]. The base mismatch was essential for the secondary structure of tRNA and played a crucial role in the accurate translation of the genetic code. It also helped minimize errors during mRNA transcription [62].

### 3.5. Non-Coding Regions

Two common non-coding regions (OL and CR) were identified in the elongate loach mitogenome, the OL region was 39 bp in length and was located between *tRNA-Asn* and *tRNA-Cys*. The CR region was located between *tRNA-Pro* and *tRNA-Phe*, which is the longest no-coding region in the entire mitochondrial genome with a span distance of 926 bp. It plays a key role in replication and transcription [63]. Similar to other vertebrates [21,64], the CR of the elongate loach exhibited the highest AT content (67.39%) among all regions of the mitochondrial genome. The palindromic sequence motifs ‘tacat’ and ‘atgta’ were related to the termination of H strand replication found in the CR of the elongate loach (Figure 7), which might complete the termination by forming a stable hairpin structure [65]. 

### 3.6. Phylogenetic Relationships

Based on the complete mitochondrial genome sequences of the elongate loach and 36 Cyprinidaes species, the phylogenetic tree was constructed. It was shown that the entire phylogenetic tree was grouped into two major clades (Figure 8). The genus Cobitis, Pangio, Triplophysa, and Acanthocobitis formed one clade and matched the subfamily Cobitinae. The Cobitis and the Pangio were sister-lineage, the Triplophysa and the Acanthocobitis were sister-lineage, and the two sister-lineages were sister-lineages to each other. The other clade consisted of Yasuhikotakia, Sinibotia, Chromobotia, Botia, Parabotia, and Leptobotia, corresponding to the subfamily Botiinae. In the subfamily Botiinae, the elongate loach was more closely related to *L. microphthalma* than to other species.

As a diverse taxa, there was a controversy in the taxonomic relationship of the subfamily Cobitinae. This study exhibited a monophyly of the subfamily Cobitinae which consists of four clades. However, according to Liu et al. [11], there were sisterhoods in many branches. Therefore, the species in Cobitinae cannot form a monophyletic group, the classification of Cobitinae in our study is incomplete, and more taxa should be used in future studies.

It is generally considered that the subfamily Botiinae is a group with a relatively clear taxonomic relationship. In this study, according to their respective genera separately, all individuals except for those of the subfamily Botiinae were clustered into a common branch, which could be confirmed as the monophyly of the subfamily Botiinae. In a previous study, the genera Botia was separated into a separate genus [7] and the genera Botia was divided into three subgenera: Sinibotia, Botia, and Hymenophysa [66]. Others did not further categorize these subgenera, but instead grouped them under the genus Botia [12,67]. In this study, subgenera Botia and subgenera Sinibotia species were clustered separately and formed parallel branches with the species of other genera. Thus, the results supported that subgenera Botia and subgenera Sinibotia should be raised to genus status. Additionally, the phylogenetic tree showed that the elongate loach and *L. microphthalma* formed a sister group which together formed a sister group of other Leptobotia species. According to Li et al. [18], the elongate loach and *L. mantschurica* were classified as sister lineages using protein genome sequencing to construct the phylogenetic tree; however, this study was analyzed based on limited taxa sampling, thus lacking sufficient phylogenetic information of the elongate loach. 

Tang et al. [13] suggested that the Leptobotia and Parabotia genera were an unnatural group and not reciprocally monophyletic groups as previously hypothesized [13,67,68,69]. They used the species “*L. mantschurica*” in the phylogenetic analysis which was nested with Parabotia and it shared the same sequences with *Parabotia mantschuricus*. However, we have not found any detailed explanation taxonomically concerning “*Leptobotia mantschurica*” and “*Parabotia mantschuricus*”. Thus, the species “Leptobotia mantschurica” is improper for use in phylogenetic analysis before clear classification. Our phylogenetic tree clearly showed that the Leptobotia and Parabotia genera were a perfect monophyly. Additionally, in the Parabotia species, part of the support value in the branch was low, suggesting that the phylogenetic relationships of these species have not been solved well. Further investigations should be performed to solve this problem.

## 4. Conclusions

In this study, we reported the complete mitogenome of the elongate loach, the structural characteristics of the mitogenome of the elongate loach were analyzed in detail, and the phylogenetic analyses of the elongate loach were inferred using the complete mitogenome. The full length of the genome sequence was 16,591 bp, and the arrangement of the elongate loach mitochondrial genome is similar to most teleost fishes. Almost all 13 PCGs showed the regular start codon ATG, except for gene *cox1*, which started with GTG. Six PCGs had incomplete stop codons T--. Thirteen PCGs were evolving under purifying selection, and the mitogenome shared a high identity with Leptobotia species. All of the tRNA genes were standard clover-shaped except for the lack of a DHU stem in *trnS1*. The phylogenetic analysis showed that the elongate loach was more closely related to *L. microphthalma* than to other species. The Leptobotia and Parabotia genera were monophyly. In this study, we first studied the selection pressure of complete PCGs in the elongate loach. Overall, we have a deeper understanding of the mitochondrial genome structure and phylogenetic analysis of the elongate loach. However, exact information about many Cobitidae fishes is still unknown. Extra taxa should be used for the phylogenetic research of Cobitidae in the future.

## Figures and Tables

**Figure 1 animals-13-03841-f001:**
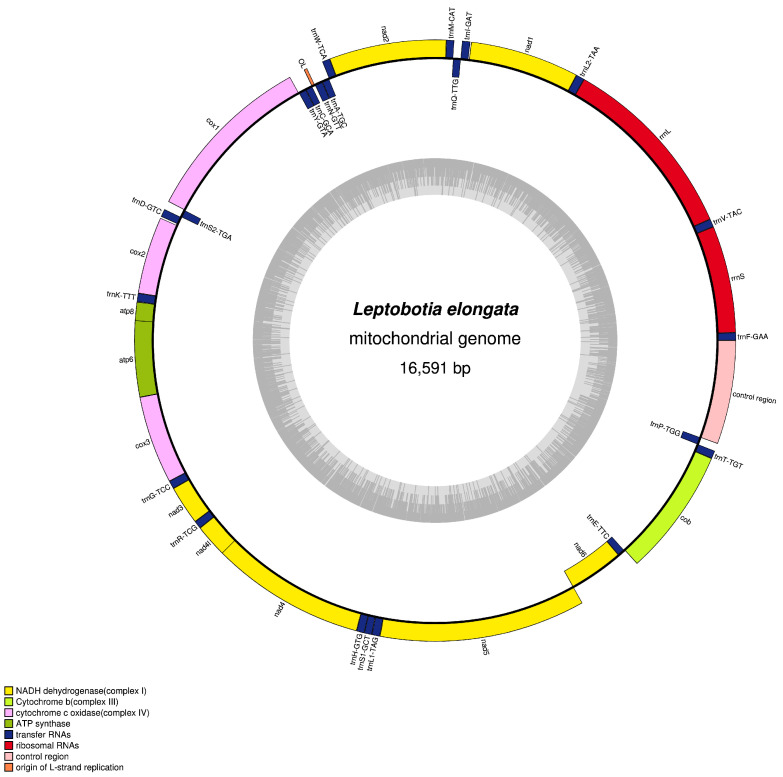
Mitochondrial genome map of the elongate loach.

**Figure 2 animals-13-03841-f002:**
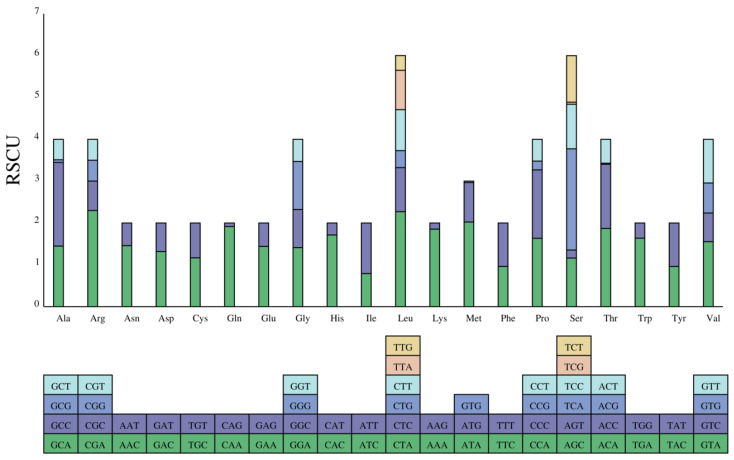
The relative synonymous codon usage (RSCU) in the mitogenome of the elongate loach.

**Figure 3 animals-13-03841-f003:**
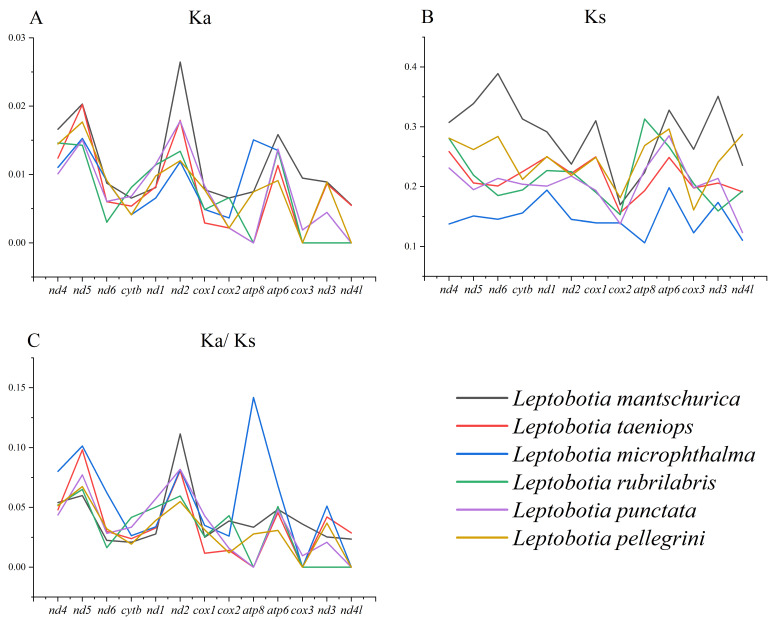
Non-synonymous (**A**) and synonymous (**B**) substitutional rates and the ratios of KaKs (**C**) of the protein coding genes of the elongate loach.

**Figure 4 animals-13-03841-f004:**
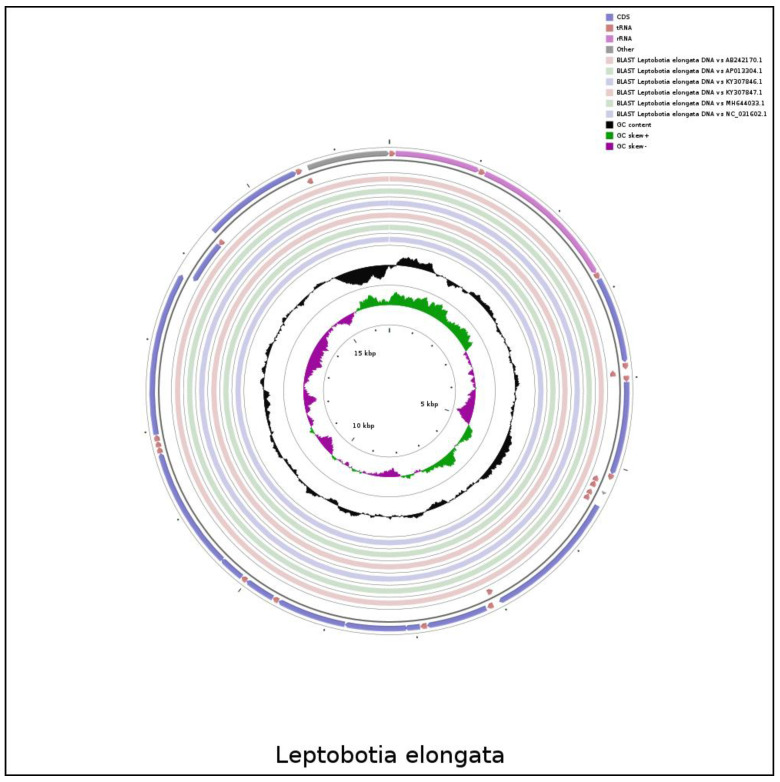
The comparative circle diagram of the genomes structure of Leptobotia species.

**Figure 5 animals-13-03841-f005:**
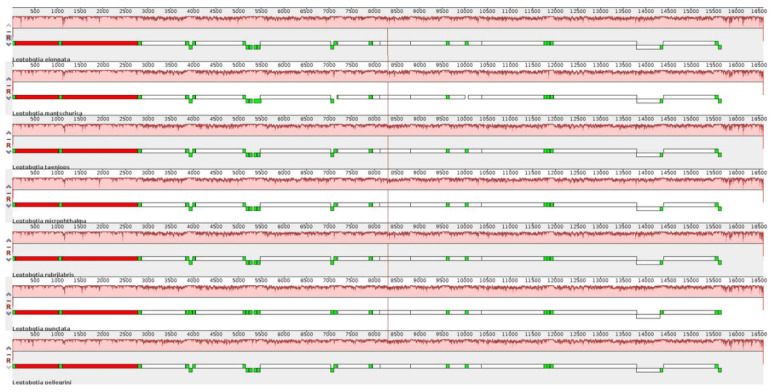
The visualized results of the genome comparison of *L. elongata*.

**Figure 6 animals-13-03841-f006:**
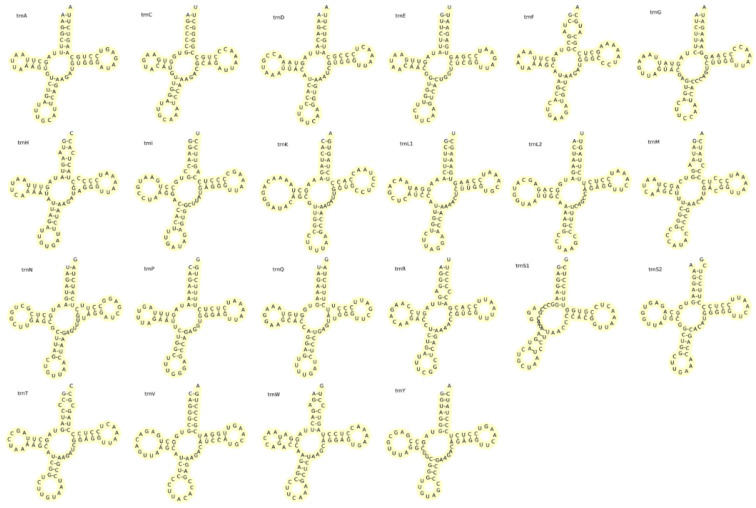
Putative secondary structure of elongate loach tRNA.

**Figure 7 animals-13-03841-f007:**
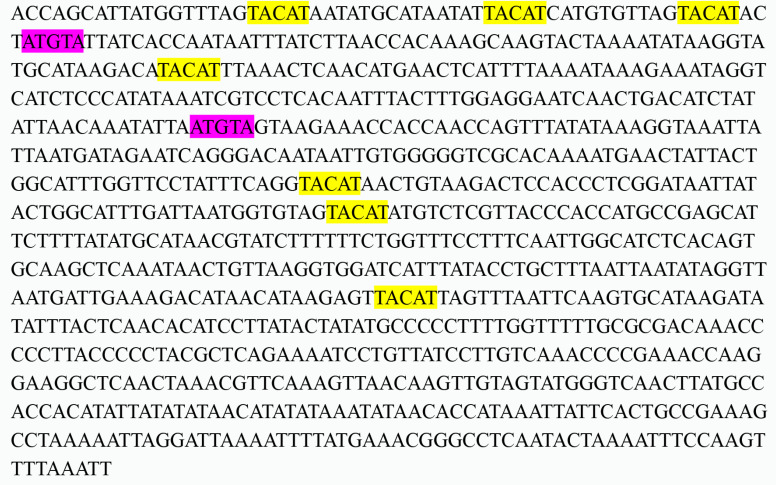
Compositional features of the control region of the elongate loach mitochondrial genome. Palindromic motif sequence ‘TACAT’ and ‘ATGTA’ are marked in yellow and purple, respectively.

**Figure 8 animals-13-03841-f008:**
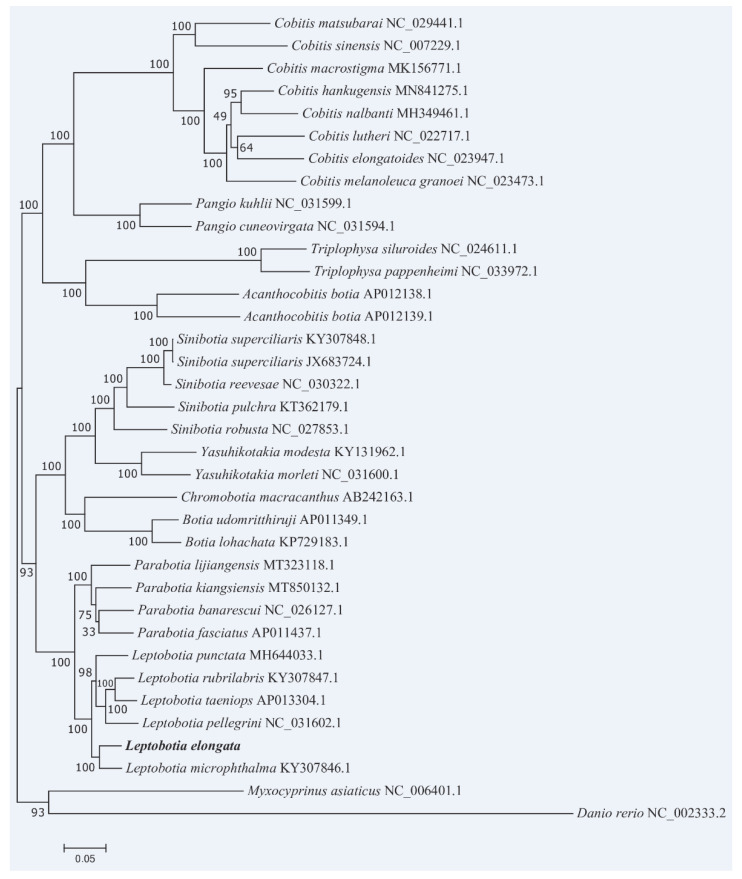
ML tree with boostrap values on the nodes constructed by using the nucleotide sequences of the 13 PCGs in the mitogenome of the elongate loach.

**Table 1 animals-13-03841-t001:** Taxonomic information and Genebank accession numbers of all species used in the phylogenetic analysis.

Family	Genus	Species	Assession Number
Myxocyprinae	Myxocyprinus	*Myxocyprinus asiaticus*	NC_006401.1
Cyprinidae	Danio	*Danio rerio*	NC_002333.2
Cobitidae	Leptobotia	*Leptobotia microphthalma*	KY307846.1
	Leptobotia	*Leptobotia elongata*	OR818399
	Leptobotia	*Leptobotia pellegrini*	NC_031602.1
	Leptobotia	*Leptobotia taeniops*	AP013304.1
	Leptobotia	*Leptobotia rubrilabris*	KY307847.1
	Leptobotia	*Leptobotia punctata*	MH644033.1
	Leptobotia	*Leptobotia mantschurica*	AB242170.1
	Parabotia	*Parabotia fasciata*	AP011437.1
	Parabotia	*Parabotia banarescui*	NC_026127.1
	Parabotia	*Parabotia kiangsiensis*	MT850132.1
	Parabotia	*Parabotia lijiangensis*	MT323118.1
	Botia	*Botia udomritthiruji*	AP011349.1
	Botia	*Botia lohachata*	KP729183.1
	Chromobotia	*Chromobotia macracanthus*	AB242163.1
	Yasuhikotakia	*Yasuhikotakia morleti*	NC_031600.1
		*Yasuhikotakia modesta*	KY131962.1
	Sinibotia	*Sinibotia robusta*	NC_027853.1
		*Sinibotia pulchra*	KT362179.1
		*Sinibotia reevesae*	NC_030322.1
		*Sinibotia superciliaris*	JX683724.1
		*Sinibotia superciliaris*	KY307848.1
	Acanthocobitis	*Acanthocobitis botia*	AP012139.1
		*Acanthocobitis botia*	AP012138.1
	Triplophysa	*Triplophysa pappenheimi*	NC_033972.1
		*Triplophysa siluroides*	NC_024611.1
	Pangio	*Pangio kuhlii*	NC_031599.1
		*Pangio cuneovirgata*	NC_031594.1
	Cobitis	*Cobitis lutheri*	NC_022717.1
		*Cobitis melanoleuca granoei*	NC_023473.1
		*Cobitis nalbanti*	MH349461.1
		*Cobitis elongatoides*	NC_023947.1
		*Cobitis hankugensis*	MN841275.1
		*Cobitis macrostigma*	MK156771.1
		*Cobitis sinensis*	NC_007229.1
		*Cobitis matsubarai*	NC_029441.1

**Table 2 animals-13-03841-t002:** Nucleotide composition and skewness values of the elongate loach mitogenome of H and L strands.

*L. elongata*	Size (bp)	A%	T%	G%	C%	A + T%	G + C%	AT-Skew	GC-Skew
Mitogenome	16,591	30.79	24.77	16.17	28.27	55.56	44.44	0.108	−0.272
PCGs	11,430	28.56	26.77	15.55	29.13	55.33	44.67	0.032	−0.304
tRNAs	1558	28.18	25.8	23.49	22.53	53.98	46.02	0.044	0.021
rRNAs	2638	34.04	19.48	21.04	25.44	53.53	46.47	0.272	−0.095
Dloop	926	35.64	31.75	13.71	18.9	67.39	32.61	0.058	−0.159

**Table 3 animals-13-03841-t003:** Summary of the elongate loach mitogenome.

		Position				Codon	
Gene	Stand	From	To	Size	Intergenic Length	Start	Stop
tRNA-*phe*	H	1	69	69	0		
12 S rRNA	H	70	1024	955	0		
tRNA-*val*	H	1025	1096	72	0		
16 S rRNA	H	1097	2779	1683	0		
tRNA-*leu*	H	2780	2854	75	0		
*nd1*	H	2855	3829	975	0	ATG	TAA
tRNA-*ile*	H	3838	3909	72	8		
tRNA-*gln*	L	3908	3978	71	−2		
tRNA-*met*	H	3980	4048	69	1		
*nd2*	H	4049	5094	1046	0	ATG	TA-
tRNA-*trp*	H	5095	5163	69	0		
tRNA-*ala*	L	5166	5234	69	2		
tRNA-*asn*	L	5236	5308	73	1		
OL	L	5310	5340	39	1		
tRNA-*cys*	L	5339	5404	66	−2		
tRNA-*tyr*	L	5406	5476	71	1		
*cox1*	H	5478	7028	1551	1	GTG	TAA
tRNA-*ser*	L	7030	7100	71	1		
tRNA-*asp*	H	7103	7174	72	2		
*cox2*	H	7188	7878	691	13	ATG	T--
tRNA-*lys*	H	7879	7954	76	0		
ATP*ase8*	H	7956	8123	168	1	ATG	TAA
ATP*ase6*	H	8114	8797	684	−10	ATG	TAA
*cox3*	H	8797	9581	785	1	ATG	TA-
tRNA-*gly*	H	9582	9653	72	0		
*nd3*	H	9654	10,002	349	0	ATG	T--
tRNA-*arg*	H	10,003	10,072	70	0		
*nd4l*	H	10,073	10,369	297	0	ATG	TAA
*nd4*	H	10,363	11,744	1382	−7	ATG	TA-
tRNA-*his*	H	11,745	11,814	70	0		
tRNA-*ser*	H	11,815	11,881	67	0		
tRNA-*leu*	H	11,883	11,955	73	1		
*nd5*	H	11,956	13,794	1839	0	ATG	TAA
*nd6*	L	13,791	14,312	522	−4	ATG	TAA
tRNA-*glu*	L	14,313	14,381	69	0		
*cytb*	H	14,386	15,526	1141	4	ATG	T--
tRNA-*thr*	H	15,527	15,598	72	0		
tRNA-*pro*	L	15,597	15,666	70	−2		
CR	H	15,666	16,591	926	0		

**Table 4 animals-13-03841-t004:** Relative synonymous codon usage and codon numbers of *L. elongata* mitochondrial PCGs.

Codon	No.	RSCU	Codon	No.	RSCU	Codon	No.	RSCU
UAA()	7	1	AAA(K)	38	1.8536	CGG(R)	5	0.5
GCA(A)	68	1.4468	AAG(K)	3	0.1464	CGU(R)	5	0.5
GCC(A)	94	2	CUA(L)	123	2.271	AGC(S)	24	1.161
GCG(A)	3	0.064	CUC(L)	57	1.0524	AGU(S)	4	0.1938
GCU(A)	23	0.4892	CUG(L)	22	0.4062	UCA(S)	50	2.4192
UGC(C)	7	1.1666	CUU(L)	53	0.9786	UCC(S)	22	1.0644
UGU(C)	5	0.8334	UUA(L)	51	0.9414	UCG(S)	1	0.0486
GAC(D)	25	1.3158	UUG(L)	19	0.351	UCU(S)	23	1.113
GAU(D)	13	0.6842	AUA(M)	60	2.0226	ACA(T)	72	1.87
GAA(E)	36	1.44	AUG(M)	28	0.9438	ACC(T)	59	1.5324
GAG(E)	14	0.56	GUG(M)	1	0.0336	ACG(T)	1	0.026
UUC(F)	63	0.9618	AAC(N)	54	1.4594	ACU(T)	22	0.5716
UUU(F)	68	1.0382	AAU(N)	20	0.5406	GUA(V)	52	1.5524
GGA(G)	48	1.4116	CCA(P)	47	1.6348	GUC(V)	23	0.6864
GGC(G)	31	0.9116	CCC(P)	47	1.6348	GUG(V)	24	0.7164
GGG(G)	39	1.1472	CCG(P)	6	0.2088	GUU(V)	35	1.0448
GGU(G)	18	0.5296	CCU(P)	15	0.5216	UGA(W)	45	1.6364
CAC(H)	42	1.7142	CAA(Q)	45	1.9148	UGG(W)	10	0.3636
CAU(H)	7	0.2858	CAG(Q)	2	0.0852	UAC(Y)	26	0.963
AUC(I)	59	0.792	CGA(R)	23	2.3	UAU(Y)	28	1.037
AUU(I)	90	1.208	CGC(R)	7	0.7			

## Data Availability

The datasets presented in this study were submitted to The National Center for Biotechnology Information (NCBI) database.

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
