# Peer review of "Characterization of the Complete Mitochondrial Genome of the Elongate Loach and Its Phylogenetic Implications in Cobitidae"

_animals, 2023, doi:10.3390/ani13243841_

Round 1
Reviewer 1 Report
Comments and Suggestions for Authors
The manuscript presents the newly obtained full mitogenome of Leptobotia elongata as a new molecular resource fot the Cobitidae family. It also includes a phylogenetic analyses with 37 Cypriniformes species.
In general, the structure is correct and the material and methods is clear, I have no concerns regarding the analyses.
However, the text needs an in-depth revision before publication, there are numerous spelling errors throughout the text, tables and figures, for example:
L. elongate instead of L. elongata
Genebank instead of GenBank
Assession instead of Accesion
Names of species not in italics
subgenra instead of subgenera
All taxa names (orders, families, subgenera...) must always begin with capital letter
Next, talking about specific cases:
Line 13: Change "phylogenetic" for "evolutionary"
Line 23: Join the two sentences, put a comma after "(PCGs)" and continue with "however, detailed information is limited"
Line 27: Delete "the" before L. elongata
Line 60: Delete "the" before L. elongata
Line 60/61: Delete "the Cobitidae family"
Line 71/72: The authors statement works only in the case of closely related or recently diverged species, but it does not work for deep phylogenies i.e. when you compare different orders.
Material and Methods: I miss a figure with a photo of the specimen, it always help the reader to get things and is also an evidence of the right species identification of the specimen.
Line 185/186: Clarigy when the authors say "species with further phylogenetic relationship". Does it mean "closely related species"?
Line 202/203: "To be polymorphism" change for "to have polymorphism"or"to be polimorphic"
Line 261: Is figure 8, not 7
Line 268: "population" is for the same species, if you are talking about a subfamily, then use "taxa"
Line 269: Delete "N"
Regarding the conclusions, I am concerned regarding how the authors have considered the sequence AB242170.1 and its position in the phylogenetic tree, mixing Leptobiota and Parabotia.
When a phylogenetic analysis is carried out and any sequence is "out of place" the very first hypothesis to be tested (specially when talking about Genbank samples) is a possible taxonomic misidentification of the sample.
I have used the COI sequence of the mitogenome used to represent L. mantschurica (AB242170.1) and the result was a 100% identification with "Parabotia mantschuricus" in BOLD systems.
With this result, I have checked the current taxonomic status of these species in "Catalog of fishes", I am not an expert in the taxonomy of Cobitidae, and looks like L. mantschurica is a synonym of Parabotia mantschuricus (or maybe the name changed?)
Therefore, there is no problem with the Parabotia clade, is perfectly monophylic.
The close relationship but independent monophyly of Parabotia and Leptobotia also agree with Tang et al. Even when the authors proclaim in Lines 296 297 that was the opposite.
Therefore, all the text about this sequence must be revised.
Comments on the Quality of English Language
I found no major concerns regarding english grammar.
Author Response
Please see the attachment. An example can be found here.

Reviewer 2 Report
Comments and Suggestions for Authors
The authors of this manuscript sequenced the mitochondrial genome of the Leptobotia elongata and characterized its structure. Additionally, they performed the phylogenetic analysis for the species with other cyprinids using the maximum likelihood method. The manuscript might be interesting to the readers of the Animals. However, it requires some revisions before publication. The major issue is the typological errors. It needs to be corrected. I have provided the detailed comments in the annotated PDF. Here, please find some major concerns.
Be careful in scientific naming. The specific name of the fish- elongata vs elongate. Italicization of scientific names.
The introduction section fails to establish the research question. Why was this study necessary? What knowledge does it add to our understanding of the loaches? These are to be added in or before the last paragraph of the introduction section.
There is some basic information about the mitochondrial DNA, its inheritance and evolution. This is not wrong but such basic info is not necessary at this level.
In the materials and methods section, there is almost nothing mentioned about the sampling methods and number of samples. How did you confirm the correct taxonomic identification of L. elongata? Need to mention it with reference.
Other comments and suggestions are in the annotated PDF. I hope they will be helpful in improving the manuscript. All the best!

There are typological errors rather than grammatical ones. So, careful revision by the authors should be enough.
Author Response
Please see the attachment.An example can be found here.

Reviewer 3 Report
Comments and Suggestions for Authors
The manuscript ‘Characterization of the complete mitochondrial genome of Leptobotia elongata and its phylogenetic implications in Cobitidae” is interesting. The knowledge gap has been clearly depicted in the manuscript and the objective of the study is very clear. Authors mentioned that although research has been done on different aspects like biological characteristics, artificial breeding etc. of L. elongate, phylogenetic relationships is limited. Therefore, the study aimed to characterize the mitogenome of the species and re-evaluate the phylogenetic relationships of L. elongate through mitogenome based approach. The authors are requested to clarify the following points.
1. How was L. elongate identified? By morphological keys? Was help of any taxonomist taken? If yes, kindly mention in acknowledgement. What was the sample size? Kindly mention.
2. The authors are requested to give more details on the mitochondrial genome assembly. What programmes used etc.
3. AT-skew and GC-skew formulas mentioned in the manuscript [ATskew=(A-T)/(AT); GCskew=(G-C)/(GC)] are not correct. It should be AT-skew=(A-T)/(A+T); GC-skew=(G-C)/(G+C). Kindly refer Perna NT, Kocher TD. Patterns of nucleotide composition at fourfold degenerate sites of animal mitochondrial genomes. J Mol Evol. 1995 Sep;41(3):353-8. doi: 10.1007/BF00186547. The authors are requested to make the necessary corrections.
4. The phylogenetic analysis part is not very clear. The authors mentioned ‘The phylogenetic tree was reconstructed using the complete mitochondrial genome ‘sequences of 37 Cypriniformes species’. All 37 genes were taken? Generally, phylogenetic analysis in mitochondrial genome is done based on concatenated nucleotide sequence of 13 protein coding genes or concatenated nucleotide sequence of 13 protein coding genes and 2 rRNAs. Kindly clarify and reconstruct the phylogenetic trees.
5. Moreover, what’s the GenBank accession number of the species? I didn’t get it in the manuscript.
Author Response

(The authors gave the same response as above.)

Round 2
Reviewer 3 Report
Comments and Suggestions for Authors
The manuscript has been improved significantly and the authors have addressed all the concerns. The revised form of the manuscript may be accepted for publication.
Comments on the Quality of English LanguageMinor issues are there.
Author Response
Cover letter
Dear Reviewers,
I sincerely appreciate your comments concerning our manuscript entitled "Characterization of the complete mitochondrial genome of Leptobotia elongata and its phylogenetic implications in Cobitidae" (animals-2709474).
The minor mistakes in our manuscript have been revised accordingly
a) The "Current studies involving phylogenetic relationships of elongate loach focused on solving inter-family relationships in Cobitidiae [6,11-13], but less research on the phylogenetic relationships between the different species" has been replaced by "Studies dealing with the phylogenic status of the elongate loach addressed questions about inter-family relationships in Cobitidae [6,11-13] but research into species phylogenetic relationship within the family remain lacking.".
b) The "polimorphic" has been revised as "polymorphic".
c) The "It used" has been replaced by "They used".
d) The " with Parabotia, it shared" has been replaced by "with Parabotia and it shared" accordingly
Thanks for your suggestion once again for revising our manuscript.